# Causal Pathway Extraction from Web-Board Documents

**Chaveevan Pechsiri [1,\*] and Rapepun Piriyakul [2]**

[1] College of Innovative Technology and Engineering, Dhurakij Pundit University, Bangkok 10210, Thailand
[2] Department of Computer Science, Ramkhamhaeng University, Bangkok 10240, Thailand; rapepunnight@yahoo.com
[\*] Correspondence: chaveevan.pec@dpu.ac.th

**Abstract:** This research aim is to extract causal pathways, particularly disease causal pathways, through cause-effect relation (CErel) extraction from web-board documents. The causal pathways benefit people with a comprehensible representation approach to disease complication. A causative/effect-concept expression is based on a verb phrase of an elementary discourse unit (EDU) or a simple sentence. The research has three main problems; how to determine CErel on an EDU-concept pair containing both causative and effect concepts in one EDU, how to extract causal pathways from EDU-concept pairs having CErel and how to indicate and represent implicit effect/causative-concept EDUs as implicit mediators with comprehension on extracted causal pathways. Therefore, we apply EDU's word co-occurrence concept (wrdCoc) as an EDU-concept and the self-Cartesian product of a wrdCoc set from the documents for extracting wrdCoc pairs having CErel into a wrdCoc-pair set from the documents after learning CErel on wrdCoc pairs by supervised-machine learning. The wrdCoc-pair set is used for extracting the causal pathways by wrdCoc-pair matching through the documents. We then propose transitive closure and a dynamic template to indicate and represent the implicit mediators with the explicit ones. In contrast to previous works, the proposed approach enables causal-pathway extraction with high accuracy from the documents.

**Keywords:** cause-effect relation; transitive closure; word co-occurrence

## 1. Introduction

The objective of this research is to extract causal pathways, particularly disease causal pathways, from downloaded disease documents from several Thai hospital web-boards. The causal pathway extraction of the research is based on determining a sequence of Cause-Effect pairs having a cause-effect relation (called 'CErel') from the documents where a Cause-Effect pair (called 'CEpair') is an ordered pair; Cause is a causative event/state concept; Effect is an effect event/state concept. According to Khoo [1], CErel is a semantic relation which is a directional link between concepts as entities that participate in the relation. Where the concepts connected by a relation are often represented as follow:

<Concept1>---(Relation)---<Concept2>

where the '<…>' and '(…)' symbols represent a concept and a relation type respectively. A dash line is a directional link between Concept1 and Concept2 and is labeled to indicate the type or meaning of the relation. With regard to our research, CErel as a cause-effect relation type is represented as follow:

<CausativeConcept>---(CErel)---<EffectConcept>

where CausativeConcept and EffectConcept are a causative concept and an effect concept respectively of either event or state occurrences on the documents. Moreover, Khoo [1]

stated that "concepts and relations are the foundation of knowledge and thought while the concepts are the building blocks of knowledge and the relations are the cement linking up the concepts into the knowledge structures," e.g., a causal chain or a causal pathway contains CausativeConcept, EffectConcept and CErel to become the knowledge structure). With regards to Staplin et al. [2], the causal pathway in epidemiologic studies is a path starting at the exposure and ending at the disease that follows the direction of the arrows. All arrows of the causal pathway point in the same direction from the exposure toward the outcome, e.g., the disease occurrence is the outcome, [3]. According to Gaskell and Sleigh [3], the causal pathway as A → $M_j$ → B contains the exposure (A) which might cause the outcome (B) or through an intermediate process or variable called a mediator ($M_j$) where $M_j$ is a single mediator if $j = 1$; and $M_j$ is either sequential-mediators or multilevel-mediators if $j = 2,3,…,$num which is an integer. However, our research concerns the extraction of the causal pathway with either a single mediator or sequential mediators mostly occurring on the documents. With regards to our disease document, A, $M_j$, and B are either a causative event/state concept or an effect event/state concept which is mostly based on a verb phase of an elementary discourse unit (EDU) where an EDU is a simple sentence or a clause [4]. The EDU expression of the research is based on the general linguistic expression in Figure 1 after stemming words and the stop word removal. Where NP1 and NP2 are noun phrases; VP is a verb phrase; Noun is a noun concept set; Verb$_{strong}$ is a strong verb concept set consisting of causative-verb concepts and effect-verb concepts; Verb$_{weak}$ is a weak verb concept set requiring more information, i.e., Noun, to become the causative/effect concept, e.g., '*have*+*fat*+*accumulate*' ('*have accumulated fat*') and '*be*+*disease*' ('*get disease*'); Adv is an adverb concept set; Adj is the adjective concept set; and Adjphrase is an adjective phrase.

EDU → NP1 VP | VP
VP → Verb NP2 | Verb adv | Verb
Verb → Verb$_{weak}$ Noun | Verb$_{strong}$
NP1 → pronoun | Noun | Noun Adj | Noun Adjphrase
NP2 → Noun | Noun Adj | Noun Adjphrase | Noun Prepphrase
Verb$_{weak}$ → {'*be*','*notBe*','*have*','*notHave*','*use, Take*'}
Verb$_{strong}$ → {'*occur*', '*constrict*', '*block up*', '*not_respond*', '*deteriorate*', '*excrete*', '*increase, enlarge*', '*change*', '*lack*', '*Response*', '*damage*', '*becomeInflamed*', '*die*', '*beStiff*', '*beThick*', '*Tear*', '*supply*', '*bleed*', '*beFailure*','*Deposit*','*flow,pass*', '*stimulus*',...}
Adj → {'*high*', ...}
Adv → {'*difficultly*', '*liquidly*', ...}
Noun → {' ', '*scar*', '*patient*', '*bloddVessel*', '*heart*', '*liver*', '*kidney*', '*myocardium*', '*brain*','*humanOrgan*', '*blood*', '*urine*', '*pressure*', '*sugar*', '*fat*', '*protein*', '*food*', '*contraction*', '*..color*', '*catalyst*', Ⅱ}

**Figure 1.** The general Thai linguistic expression of EDU after stemming words and eliminating stop words.

The example of a causal pathway is expressed on the document by a sequence of Cause-Effect pairs (CEpairs) having CErel on the document is shown in Example 1.

**Example 1.** *Topic Name:* *How important is the hormone insulin for the body?*
　*(From Bangkok Hospital Phuket) …*
*EDU1.*　　"*When a patient lacks insulin,*"
　　　　(*When* (*a patient*)/NP1 ((*lacks*)/Verb$_{strong}$ (*insulin*)/NP2)/VP
*EDU2.*　　"
　　　　*causing the body to be unable to use sugar as energy in various parts of the body*".
　　　　*causing* ((*the body*)/NP1
　　　　(*is unable* (*to take*)/Verb$_{strong}$ (*sugar to use as energy in various part of body*)/NP2)/VP

*EDU3.* 　　*"[Thai]/**Causing [the patient] to have hyperglycaemia**".*
　　　　　*[Thai]/**causing** ([Thai]/**the patient**])/NP1*
　　　　　*(([Thai]/**has**)/Verb<sub>weak</sub> ([Thai]/**hyperglycaemia**)/NP2)/VP*

*EDU4.* 　　*"[Thai]/**And causing [the patient] to be diabetes**". .........*
　　　　　*[Thai]/**and causing** ([Thai]/**the patient**])/NP1*
　　　　　*(([Thai]/**gets**)/Verb<sub>weak</sub> ([Thai]/**diabetes**)/NP2)/VP*

*where the [..] symbol means a term/terms inside the symbol being ellipsis; and a '[Thai]/causing' term is a causal verb with the part of speech as a causal conjunction between a causative-concept EDU and an effect-concept EDU.*

　　Example 1 shows a sequence of CEpairs having CErel on an EDU-pair sequence as shown in the following expression.

　　EDU1-EDU2 Pair as the first CEpair (CEpair₁): EDU1<Cause>---(CErel)---><Effect>EDU2
　　EDU2-EDU3 Pair as the second CEpair (CEpair₂): EDU2<Cause>---CErel---><Effect>EDU3
　　EDU3-EDU4 Pair as the third CEpair (CEpair₃): EDU3 <Cause>---(CErel)---><Effect>EDU4

　　The sequence of CEpair₁ through CEpair₃ is the causal pathway as EDU1→EDU2→EDU3→EDU4.

　　According to Example 1, EDU1 (a causative-concept EDU), and EDU4 (an effect-concept EDU) are the exposure and the outcome respectively whilst EDU2 and EDU3 (which are effect/causative-concept EDUs) are sequential-mediators.

　　The extracted causal pathways would support the problem-solving system by supporting unprofessional persons to have a more comprehensible approach to the disease complication through social media which results in compliance to the physician's suggestion of the appropriate treatment. Therefore, the research concerns extracting the causal pathways represented by the CEpair*ᵢ* sequences (where *i* is an index of a CEpair in a sequence) from the document.

　　There are several techniques [5–13] having been applied for determining or extracting the causal pathways or the causal chains through the cause-effect/causal relation determination between two entities/events from the documents (see Section 2). The features used for the CErel determination from the previous research are mainly version a noun variable pair or a verb variable pair from a noun phrase pair or a verb phrase pair respectively within one simple sentence or a simple sentence pair. Whilst the causative/effect concepts of the events/states on the Thai documents are mostly based on the EDUs' verb phrase expressions where the same verb phrase expressions with the different NP1 concepts have the different causative/effect concepts of the events/states, e.g., EDU1: "([Thai]/*Blood clots*)NP1 ([Thai]/*flow in the artery*)/VP" and EDU2:"([Thai]/Fat)/NP1 ([Thai]/*flow in the artery*)/VP" have the *flow*(*BloodClot*, *artery*) and *flow*(*Fat*, *artery*) concepts respectively after stemming words and stop words removal. Therefore, the features used for the CErel determination of our research are based on composite variables relied on a predicate-argument term set for obtaining a causative/effect concept. Where a composite variable is a variable made up of two or more individual variables, called indicators, into a single variable [14]. Each indicator alone doesn't provide sufficient information, but altogether they can represent the more complex concept. In addition, the entailment classification of the previous research [15] is based on the similarity scores which cannot apply to our disease documents, e.g., the relation between EDU1: "[Thai]/*The patient get arteriosclerosis*" and EDU2: "[Thai]/*because fat deposits on the artery wall*". is CErel with the similarity score approaching zero. Moreover, the actual causal pathway determination from texts of the previous research are mostly based on two steps of the cause-effect relation type, e.g., A causes B and B causes C, without concerning the implicit mediator whereas our disease documents contain several steps or more than two steps of the cause-effect relation type including the implicit mediators. With regard to the causal pathway for the problem-solving system, the implicit mediators on the causal pathway should be represented by the explicit mediators to have the complete causal pathway for understanding the mechanism through which the composite variable affects the outcome.

However, the Thai documents have several specific characteristics, such as zero anaphora or the implicit noun phrase, without word and sentence delimiters, etc. All of these characteristics are involved in three main problems (see Section 3): (1) how to determine CErel on each EDU-concept pair from the documents where there are some EDU occurrences with both the causative concept in one CErel and an effect concept in another CErel; (2) how to extract causal pathways from several EDU-concept pairs having CErel; and (3) how to indicate the implicit mediators or the implicit effect/causative-concept EDUs on the correct extracted causal pathways from the documents for representing the implicit mediators in the form of the explicit effect/causative-concept EDUs or the explicit mediators for clear comprehension. Regarding these three main problems, we need to develop a framework which combines machine learning and the linguistic phenomena to represent each EDU occurrence by an EDU's word co-occurrence (called 'wrdCo') based on wrdCoPattern on Equation (1) relying on a predicate argument pattern of an EDU occurrence (see Figure 1) after stemming words and eliminating stop words. In addition, each EDU concept of an event/state is represented by an EDU's wrdCo concept (called 'wrdCoc') as a feature or an element of a wrdCo concept set or a wrdCoc set (WC).

$$wrdCoPattern = V + W1 + W2 \tag{1}$$

where V is a predicate verb set; $V = Verb_{strong} \cup V_{inf}$; $v_a \in V$ ; Each element of $V_{inf}$ consists of $v_{weak,b} + w_{inf,c}$ ($v_{weak,b} \in Verb_{weak}$, $w_{inf,c} \in Noun$, and $w_{inf,c}$ is a word right after $v_{weak,b}$; *a*, *b*, *c*, *d*, and *e* are an integer.;

W1 is an agent argument set; $w_{1,d} \in W1$; $w_{1,d}$ is a head noun or a Noun element of NP1 and $w_{1,d}$ is a Noun element of the previous EDU's NP1 if the current EDU's NP1 is ellipsis;

W2 is a linguistic patient/information set; $w_{2,e} \in W2$ ; $W2 = Noun \cup Adv \cup Adj$ and $w_{2,e}$ is also a word sequence right after $v_a$ ; $w_{2,e}$ has a null value if $w_{2,e}$ doesn't exist;

And all $Verb_{strong}$, $Verb_{weak}$, Noun, Adv, Adj sets are based on Figure 1)

Likewise, three contributions of this paper are statistically-based approaches involved with linguistic phenomena and machine learning. The first one is that each wrdCoc feature used for the CErel determination by machine learning is the composite-variable consisting of the elements of V, W1, and W2 for the causative concept/effect concept representation. The second one is that our extracted causal pathways contain more than two steps of the cause-effect relation type and are the actual causal pathways from the documents. And the third one is that some extracted causal pathways contain the implicit mediators (or the implicit effect/causative-concept EDUs) as the implicit wrdCoc features which have to be represented by the explicit wedCoc features for clear comprehensible pathways. Moreover, our implicit wrdCoc features the qualitative data whereas the previous research [16] discovered the hidden semantics or the latent semantics as the implicit features by the graph regularization where the latent semantics of [16] is the quantitative data.

We then apply the self-Cartesian product of WC × WC [17] (the first WC as the causative-concept set, the second WC as the effect-concept set) to a test corpus for extracting wrdCoc pairs having CErel into WCP (WCP is a set of wrdCoc pairs having CErel) after learning CErel on wrdCoc pairs by naïve Bayes (NB) [18], support vector machine (SVM) [19], and logistic regression (LR) [20] from a learning corpus. According to the test corpus, all WC elements are determined by wrdCo-expression matching between wrdCo expressions of the test corpus and wrdCo expressions of the semi-automatic annotated corpus having annotated wrdCoc features. WCP is used for extracting the causal pathways through wrdCoc-pair matching on the documents (see Sections 3.1 and 3.2). We then propose using transitive closure of a binary relation over a causative concept set and an effect concept set [21] to indicate the implicit mediator occurrences on the correct extracted causal pathways and also using a dynamic template to collect the correct extracted causal pathways with the explicit mediators used for

representing those implicit mediators (where the explicit mediators are the explicit effect/causative-concept EDUs represented by EDUs'wrdCoc features) (see Section 3.3).

Our research is organized into six sections. In Section 2, related work is summarized. Problems in the causal-pathway extraction from the documents are described in Section 3 and Section 4 shows our framework for the causal pathway extraction from the documents. In Section 5, we evaluate our proposed model including discussion and then present a conclusion in Section 6.

## 2. Related Works

Several strategies [5–13] have been proposed to determine/extract a causal pathway, a causal chain, or causal path of a graph/network through the cause-effect/causal relation determination except [13] working on the implicit knowledge completion where [5,6] working on only the causal/cause-effect relation determination from texts. Girju [5] proposed decision-tree learning the causal relation from a sentence based on the lexico-syntactic pattern (NP1 causal-verb NP2) where NP1 is a cause and NP2 is an effect or vice versa. Cao et al. [6] also used syntactic patterns by manually annotating one sentence or between two sentences having a cue (a word or a phrase) as a cause-effect link to express the cause-effect relation which is the core of scientific papers. Their cause-effect links were extracted by a syntactic pattern-based algorithm from scientific papers with 47% and 70% on average precision and recall respectively. Chang and Choi [7] extracted causality/causal relation with an F-score of 77.37% based on one complex sentence or two simple sentences by using a cue-phrase set to connect two noun phrases (or an NP pair) as a cause and an effect including probabilities. The extracted causal relations were used for constructing the causal paths of the causal network for the term protein having two relations; the causal relation and the hypernym relation. Pechsiri and Piriyakul [8] applied verb-pair rules resulting from machine learning techniques to extract the causality or the cause-effect relation from several simple sentences to construct one cause with several effect paths on an explanation knowledge graph. The cause-effect paths of [8] were emphasized on the consequence or concurrent occurrence of the extracted effect events. Whilst a causal chain [9] was generated by connecting the extracted causal relations with sentence's word similarity and topic matching between a causative sentence of one causal relation and an effect sentence of another causal relation where the causal relation extraction was based on clue words. Kang et al. [10] applied the Granger causality model with features, i.e., N-words, topics, sentiments, etc., to detect cause-effect relationships from texts for a time series. And [10] also applied a neural reasoning algorithm based on human annotation along with BLEU (bilingual evaluation understudy) scores used for measuring the connection of two cause-effect relationships to construct a causal chain with 57% accuracy based on expert judgments. However, the cause/effect events or entities [10] are mainly expressed by noun phrases based on day-by-day time series. Izumi and Sakaji [11] applied a causal verb set as the edge/relation to construct causal paths by connecting between a cause node and an effect node expressed by noun phrases within one sentence. The causal chain was constructed by manually selecting word vector similarities between effect nodes and cause nodes from different causal relations. Nordon et al. [12] extracted several causal relations based on the lexico-syntactic pattern [5], and then applied the text analysis, i.e., word co-occurrence and Word2vec, to determine the edge weights for solving each causal path of the causal graph from the extracted causal relations. Moreover, Ref. [13] applied the similarity score between two word-pairs as an event pair including the notified event location to calculate the event relevant for automatically discovering implicit event knowledge occurring among the sequential event chain of actions from a Japanese web blog corpus without the CErel consideration between the event pair, e.g., "roll on the floor", "sitting on a sofa", and "drinking tea" were the event chain of actions with the Living room location added. [13] evaluated the knowledge completion for the chain of actions (including the notified event location) by the graduate students scoring as 3.0 based on a five-point Likert scale.

Therefore, the causal relation determination of the previous research [5–12] is mostly based on noun phrases within one or two simple sentences except [8] using only verb pairs to extract the causal relation from several simple sentences. However, CErel of our research is based on wrdCoPattern on Equation (1) included an NP1 head noun and an EDU's verb phrase because the different agents (NP1) with the same predicate verb provide the different semantics of causative/effect concepts. The causal pathways, the causal chain, or the causal-graph paths of the previous research [7–12] are determined/extracted from documents without concerning the implicit mediator on the certain path/chain whilst [13] emphasizes the implicit knowledge completion on the event chain of actions without the CErel consideration. However, there are a few works on extracting the causal pathway from texts with little concerning in the implicit mediator.

## 3. Problems of Causal-Pathway Extraction

### 3.1. How to Determine CErel on an EDU-Concept Pair/a wrdCoc Pair

According to the corpus behavior study of the medical care domain, most of the causative/effect-concept EDUs are the events or states expressed by verb phrases. There are some verb phrase expressions with both the causative concepts and the effect concepts on the documents as shown in Example 1, e.g., EDU2 is an effect-event concept and a causative-event concept for CEpair1 and CEpair2 respectively where CEpair1 and CEpair2 are consecutive. Moreover, lack of the sentence delimiter in the Thai documents causes a problem of determining EDU's concept pairs (e.g., an EDU1-EDU2 pair or an EDU2-EDU3 pair) having CErel from three consecutive EDUs if the second EDU contains a discourse-marker cue set, {'เพราะ/*because*', 'เนื่องจาก/*since*', …}, as shown in Example 2. Where each EDU concept is represented by wrdCoc, i.e., an EDU$_j$ concept is represented by EDU$_j$'s wrdCoc called wrdCoc$_{EDUj}$, $j$ is an integer.

**Example 2.** *Topic Name:* เบาหวาน/*Diabetes*…

*EDU1.* "ผู้ป่วยเบาหวานอาจเป็นโรคหัวใจ/*A diabetic patient might get heart disease*".
(ผู้ป่วยเบาหวาน/*A diabetic patient*)/NP1
((อาจเป็น/*might get*)/Verb$_{weak}$ (โรคหัวใจ/*the heart disease*)/NP2)/VP
wrdCoc$_{EDU1}$ = getHeartDisease(person)

*EDU2.* "เนื่องจาก [ผู้ป่วย] มีน้ำตาลในเลือดสูง/*Since [the patient] has hyperglycaemia,*"
เนื่องจาก/*Since* ([ผู้ป่วย/*the patient*])/NP1
((มี/*has*)/Verb$_{weak}$ น้ำตาลในเลือดสูง/*hyperglycaemia*)/VP
wrdCoc$_{EDU2}$ = haveHyperGlycaemia(person)

*EDU3.* EDU3:"ทำให้สารเคมีบางตัวในเลือดสูงขึ้น/*causing some chemicals increase in the blood.* "
…
ทำให้/*causing* (สารเคมีบางตัว/*some chemicals*)/NP1
((สูงขึ้น/*increase*)/Verb$_{strong}$ ในเลือด/*the blood* )/VP
wrdCoc$_{EDU3}$ = increase(chemical,blood)

Example 2 contains a CEpair$_i$ with CErel as shown in the following:
wrdCoc$_{EDU2}$-wrdCoc$_{EDU3}$ Pair asCEpair1:wrdCoc$_{EDU2}$<Cause>---(CErel)---><Effect> wrdCoc$_{EDU3}$.

Therefore, we apply the self-Cartesian product of WC × WC having the first WC as a causative concept set and the second WC as an effect concept set to the test corpus for extracting the wrdCoc pairs of an EDU pairs having CErel into WCP after learning CErel on each wrdCoc pair by NB, SVM, and LR from the learning corpus (see Section 4.2). The WC elements are collected by the wrdCo-expression matching between the wrdCo expressions of the test corpus and the wrdCo expressions of the learning corpus with the semi-automatic annotated wrdCoc features (see Section 4.1).

*3.2. How to Extract the Causal Pathways*

During the causal pathway extraction, some causal pathways mingle with non-causative/effect concept EDU(s) and remain a challenge, e.g., Example 1 mingles with a non-causative/effect concept EDU(s) such as EDU2: "อินซูลินมีหน้าที่ส่งสัญญาณไปเซลล์เอาน้ำตาลไปใช้/*Insulin has a function of signaling cells to take sugar for use*". which intervenes after EDU1 of Example 1 as shown in the following:

EDU1.　"เมื่อผู้ป่วยขาดอินซูลิน/**When a patient lacks insulin,**"

　　　wrdCoc$_{EDU1}$ = lack(person,insulin)

EDU2.　"อินซูลินมีหน้าที่ส่งสัญญาณไปเซลล์เอาน้ำตาลไปใช้/**Insulin has a function of signaling cells to take sugar for use**".

(อินซูลิน/**Insulin**)/NP1

((มี/**has**)/Verb$_{weak}$ หน้าที่/**a function of** ส่งสัญญาณไปเซลล์/**signaling cells**

เอาน้ำตาล/**to take sugar** ไปใช้/**for use**)/VP

wrdCoc$_{EDU3}$ = hasFunction(insulin,signaling)

EDU3.　"ทำให้ร่างกายไม่สามารถนำน้ำตาลไปใช้เป็นพลังงานตามส่วนต่างๆของร่างกาย/**causing the body to be unable to use sugar as energy in various parts of the body**".

wrdCoc$_{EDU1}$ = beUnableToUseSugar(person)

EDU4.　"ทำให้[ผู้ป่วย]มีน้ำตาลในเลือดสูง/**Causing [the patient] to have hyperglycaemia**".

wrdCoc$_{EDU1}$ = haveHyperglycaemia(person)

EDU5.　"และทำให้[ผู้ป่วย]เป็นเบาหวาน/**And causing [the patient] to be diabetes**". .........

wrdCoc$_{EDU1}$ = getDiabetes(person)

Each wrdCoc feature having a predicate verb $v_a \in$ Verb$_{strong} \cup$ V$_{inf}$ on the test-corpus document is sequentially collected into an array of wrdCoc features for the causal pathway extraction. Where all wrdCoc features of the test-corpus document are obtained by the wrdCo-expression matching between the wrdCo expressions of the test-corpus document and the wrdCo expressions of the annotated corpus having the annotated wrdCoc features.

Therefore, we apply WCP to extract each causal pathway by the wrdCoc-pair matching on sliding window size of two consecutive wrdCoc features (or a wrdCoc pair) on the array of wrdCoc features to match among WCP elements with one wrdCoc distance through the array. If there is no match on wrdCoc-pair matching, we will stop sliding the window and then obtain a causal pathway (Section 4.4).

*3.3. How to Indicate Implicit Mediators for Explicit Mediator Representation*

Some determined causal pathways contain the implicit mediators (the implicit effect/causative-concept EDUs) as in Examples 3–4 of the same disease group.

**Example 3.** *Topic Name:* เบาหวานลงไต *(โรงพยาบาลวิภาวดี)/Diabetic Nephropathy (from Vibhavadi Hospital)*

*EDU1.　"ผู้ป่วยเป็นเบาหวานมาหลายปี/A Patient gets a diabetic disease for several years".)*

*(ผู้ป่วย/A patient)/NP1*

*((เป็น/get)/Verb$_{weak}$ เบาหวาน/a diabetes มาหลายปี/for several years)/VP*

*wrdCoc$_{EDU1}$ = getDiabetes(person)*

*EDU2.　"เนื่องจาก[ผู้ป่วย]มีน้ำตาลในเลือดสูงเป็นเวลานาน/*

*Since [the patient] have hyperglycaemia for a long period of time, "*

*เนื่องจาก/Since ([ผู้ป่วย/the patient])/NP1*

*((มี/has)/Verb$_{weak}$ น้ำตาลในเลือดสูง/hyperglycaemia*

*เป็นเวลานาน/for a long period of time)/NP2)/VP*

*wrdCoc$_{EDU2}$ = haveHyperglycaemia(person,long-time)*

*EDU3.　"ทำให้หลอดเลือดทั่วร่างกายแข็ง/causing blood vessels of whole body to be stiff and thick".*

*ทำให้/causing (หลอดเลือดทั่วร่างกาย/blood vessels of whole body)/NP1*

*(จะ/will แข็ง/be stiff)/Verb$_{strong}$ และหนา/and thick)/Verb$_{strong}$)/VP*

*wrdCoc$_{EDU3}$ = beStiff&Thick(bloodVessel)*

*EDU4.*       "ᶜᵃᵘⁿⱼ฿ᵉᵗᵃᶜᵉ/
*Causing the blood supply less to the parts of the body"*.
ᶜᵃᵒⁿ/*causing (ᵉᶩᵉ/blood)/NP1*⫶
*((ᵗ⁻ᵉᶩᵉ/supplies)/Verb$_{strong}$*⫶*less* ûᶜᵃᶜᵉᵉᴬˆᴬᵃᶦᵉᶩᵉ/*to the parts of the body)/VP*
*wrdCoc$_{EDU4}$ = beSupplied(blood,less)*

*EDU5.*       "ᶜᵉᴱᶩûⁿ[ᴱᴺᴬᵉ] ᵃûᴬˇᵡᶻ/*Causing [the patient] gets a chronic kidney disease"*. ….
ᶜᵉᴱᶩûⁿ/*causing ([ᴱᴺᴬᵉ/the patient])/NP1*
*(ᵃû/gets)/Verb$_{weak}$ (ᴬˇᵡᶻ/kidney disease)/NP2)/VP*
*wrdCoc$_{EDU5}$ = getKidneyDisease(person)*

Example 3 shows a sequence of CEpairs having CErel as follow:
wrdCoc$_{EDU2}$-wrdCoc$_{EDU1}$ Pair as CEpair$_1$: wrdCoc$_{EDU2}$<Cause>---(CErel)---><Effect> wrdCoc$_{EDU1}$
wrdCoc$_{EDU2}$-wrdCoc$_{EDU3}$ Pair as CEpair$_2$: wrdCoc$_{EDU2}$<Cause>---(CErel)---><Effect> wrdCoc$_{EDU3}$
wrdCoc$_{EDU3}$-wrdCoc$_{EDU4}$ Pair as CEpair$_3$: wrdCoc$_{EDU3}$<Cause>---(CErel)---><Effect> wrdCoc$_{EDU4}$
wrdCoc$_{EDU4}$-wrdCoc$_{EDU5}$ Pair as CEpair$_4$: wrdCoc$_{EDU4}$<Cause>---(CErel)---><Effect> wrdCoc$_{EDU5}$
CEpair$_1$ (EDU1–EDU2) is the first causal pathway expression and CEpair$_2$–CEpair$_4$
(EDU2–EDU5) are the second causal pathway expression as show in Figure 2.

**Figure 2.** Show causal pathway expressions of Example 3.

**Example 4.** *TopicName:* ᴬˇ½ᵃᴬᵃ⫶ᴬˇᵡᵈ*Diabetes and kidney disease* (from Sukumvit Hospital)
…

*EDU1.*       "ᶜᵃᴱᵗᵉᶜᵃᵒ|ûᶩᵃ°Çᵗᵉᵗᵃîᶦᴱᴱᴬ|ᵒᵒᵒ/*If hyperglycaemia occurs for a long-term,"*
ᵒᵉ/*If (ᴱᵗᵉᶜᵃᵒ|ûᶩᵃ°Çᵉ/Hyperglycaemia)/NP1*
*((ᵗᵉᵗᵃ/occurs)/Verb$_{strong}$ ᵃîᶦᴱᴱᴬ|ᵒᵒᵒ/for a long-term)/VP*
*wrdCoc$_{EDU1}$ = occur(hyperglycaemia,long-term)*

*EDU2.*        "ᴱᵒᵉᶩⁿ|ᴬ°ᵃᶩᵉᵗᴱᴬˇ︑/*the vascular wall will become inflamed"*.
*(ᴱᵒᵉᶩⁿ|ᴬ°ᵃᶩᵉ/The vascular wall)/NP1*
*(ᴱᵉ/will (ᴬˇ︑/become inflamed)/Verb$_{strong}$)/VP*
*wrdCoc$_{EDU2}$ = becomeInflamed(bloodVesselWall)*

*EDU3.*        "ᶜᵃᵒᶩⁿ|ᴬ°ᵃᶩᵉᵘˆᶦᶩᴱᵒᴬ/*Causing the arteries to be stiff and narrow"*.
ᶜᵃᵒᶩⁿ/*causing (ᵑ|ᴬ°ᵃᶩᵉ/the arteries)/NP1*
*((ᵘˆᵈ/be stiff)/Verb$_{strong}$ ᵘ|ᴱand⫶ᴳᴬ/be narrow)/Verb$_{strong}$)/VP*
*wrdCoc$_{EDU3}$ = beStiff&Narrow(bloodVessel)*

*EDU4.*        "ᵒᵗᵘᴱⁿ|ᴬ°ᵃᶩᵉᵃˆᵗ⫶⫶⫶ᴬ°ᵃᶩᵉᵡᵒᶦᵗᴱᵗᵉᵗᴱᵉ|﹐ᴱᵒᵢ̧ᴬᵒ/
*Then, small blood vessels such as renal arteries are often affected first"*.
ᵒᵗᵘᴱⁿ/*Then (ᵑ|ᴬ°ᵃᶩᵉᵃᵗᴬ/small blood vessels*
*ᴱᵃᵑ|ᴬ°ᵃᶩᵉᵡᶻ/such as renal arteries)/NP1*
*(ᵢᶴᴱᴱ/often (ᵡᵒᴱᵡ/gets)Verb$_{weak}$ ᴱ|﹐ᴱᵒᵢ̧ᴬᵒ/affected first)/VP*
*wrdCoc$_{EDU4}$ = getAffect(bloodVessel)*

*EDU5.*        "ᶜᵃᵒᶩⁿᴱᴺᴬᴅᵗᵉ́ᵒᴱᵡᴬᵒ/*Causing the patient to have kidney failure"*. …
ᶜᵃᵒᶩⁿ/*causing (ᴱᴺᴬᵉ/the patient)/NP1*
*((ᵗᵉ́/have)/Verb$_{weak}$ ᵉᴬᴱᵡᴬᵒ/kidneyfailure)/VP*
*wrdCoc$_{EDU5}$ = haveKidneyFailure(person)*

Example 4 shows a sequence of CEpairs having CErel as follow:
wrdCoc$_{EDU1}$-wrdCoc$_{EDU2}$ Pair as CEpair$_1$: wrdCoc$_{EDU1}$<Cause>---(CErel)---><Effect> wrdCoc$_{EDU2}$
wrdCoc$_{EDU2}$-wrdCoc$_{EDU3}$ Pair as CEpair$_2$: wrdCoc$_{EDU2}$<Cause>---(CErel)---><Effect> wrdCoc$_{EDU3}$
wrdCoc$_{EDU3}$-wrdCoc$_{EDU5}$ Pair as CEpair$_3$: wrdCoc$_{EDU3}$<Cause>---(CErel)---><Effect> wrdCoc$_{EDU5}$

According to Example 4, the causal pathway of Figure 3, particularly in a dash-line square, contains EDU2 as an implicit mediator between EDU2 and EDU3 in another dash-line square of the causal pathway in Figure 2. Whilst EDU4 of the causal pathway in Figure 2 is another implicit mediator between EDU3 and EDU5 of the causal pathway in Figure 3. Where the chronic kidney disease and the kidney failure have the same concept of the kidney deterioration. Therefore, we propose using TransCEPair (which is a set of CEpairs having CErel to be transitive (which is equivalent to an implicit mediator) from Transitive Closure of the binary relation over all correct extracted causal pathways) to indicate the implicit mediators on each correct extracted causal pathway and using the following ExplicitCEpairWithCErelPathways template as the dynamic template to collect the extracted causal pathways with the explicit mediators represented by EDUs' wrdCoc features used for representing the implicit mediators (see Section 4.5).

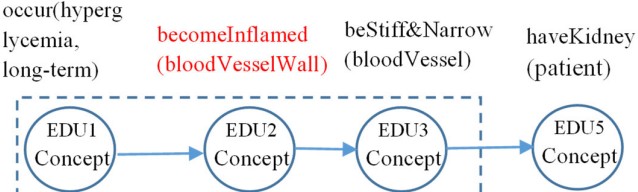

**Figure 3.** Show a causal pathway expression (EDU1, EDU2, EDU3 and EDU5) of Example 4.

**Dynamic ExplicitCEpairWithCErelPathways Template:**

$\text{wrdCoc}_{\text{EDU}j}$ -$\text{wrdCoc}_{\text{EDU}j+1}$ Pair as $\text{CEpair}_{p1}$:$\text{wrdCoc}_{\text{EDU}j}$<Cause>---(CErel)---><Effect>$\text{wrdCoc}_{\text{EDU}j+1}$
$\text{wrdCoc}_{\text{EDU}j+1}$-$\text{wrdCoc}_{\text{EDU}j+2}$Pair as $\text{CEpair}_{p2}$:$\text{wrdCoc}_{\text{EDU}j+1}$<Cause>---(CErel)---><Effect>$\text{wrdCoc}_{\text{EDU}j+2}$

… … … … …

$\text{wrdCoc}_{\text{EDU}j+n-1}$-$\text{wrdCoc}_{\text{EDU}j+n}$ Pair as $\text{CEpair}_{pn}$:$\text{wrdCoc}_{\text{EDU}j+n-1}$<Cause>---(CErel)---><Effect>$\text{wrdCoc}_{\text{EDU}j+n}$

$(\text{CEpair}_{p1} \text{ CEpair}_{p2}… \text{ CEpair}_{pn})_p$: **ExplicitCEpairWithCErelPathways**

where $p$ is a causal-pathway number; $p$ = 1, 2, …$m$; $m$, $j$, $n$, are an integer; $\text{EDU}_{j+t} \Leftrightarrow \text{EDU}_{j+t+1}$; $t$ = 0, 1, 2, …, $n-1$.

In addition to TransCEPair, the TransCEPair elements are collected by calculating the transitive closure of the binary relation [21,22] (see Figure 4) linking each node ($c_{j+t}$) having a Cause/causative-concept (where $c_{j+t}$ is represented by $\text{wrdCoc}_{\text{EDU}j+t}$ ; $j$ = 1;$t$ = 0, 1, 2, …, $n-1$; $n$ is the number of nodes) to an $e_{j+t+s}$ node having Effect/effect-concept (where $e_{j+t+s}$ represented by $\text{wrdCoc}_{\text{EDU}j+t+s}$ ; $s$ = 2, 3, … $n-1$) on the correct extracted causal pathways (Equation (2)).

$$TransCEPair \; = \bigcup_{k=1}^{numberCP} \{c_1 e_3, c_1 e_4,.., c_1 e_{n\_k}, c_2 e_4,.., c_2 e_{n\_k},.., c_{(n\_k)-2}\; e_{n\_k}\}$$

(2)

where *numberCP* is the number of correct extracted causal pathways.

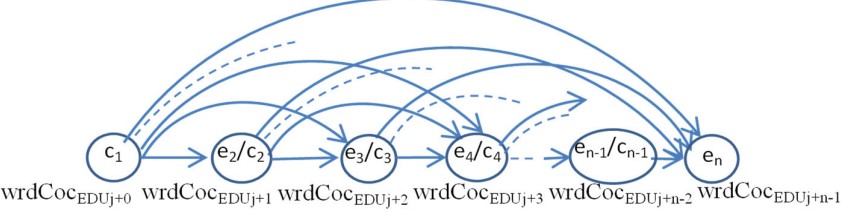

**Figure 4.** Apply the transitive closure on a correct extracted causal pathway started with $c_{j+t} = c_1$ where $j$ = 1; $t$ = 0, 1, 2, …, $n-1$.

## 4. System Overview

There are five steps in our framework, Corpus Preparation, CErel Learning on Each wrdCoc Pair, Determination of wrdCoc Pairs Having CErel, Causal Pathway Extraction, and Implicit-Mediator Indication and Representation with Explicit-Mediator from Dynamic Template as shown in Figure 5.

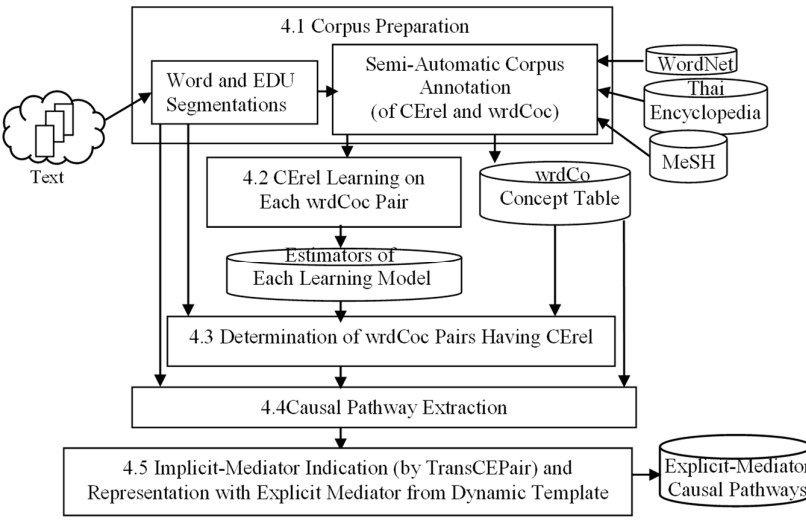

**Figure 5.** System overview.

### 4.1. Corpus Preparation.

#### 4.1.1. Word and EDU Segmentations

This step is to prepare an EDU corpus from disease-explanation documents downloaded from several hospital web-boards (http://haamor.com/;http://www.bangkok health.com; http://www.si.mahidol.ac.th/sidoctor/e-pl/; https://www.bumrungrad.com; etc. accessed on 10 August 2021). The step involves using Thai word-segmentation tools [23] and Named-Entity recognition [24,25]. After the word segmentation is achieved, EDU Segmentation [26,27] is then operated to provide an 8000 EDU corpus (consists of 4000 EDUs from a diabetes and kidney disease group and 4000 EDUs from a heart and artery disease group). This 8000 EDUs' corpus is separated into 2 parts after stemming words and the stop word removal. The first part (which consists of 2000 EDUs from the diabetes and kidney disease group and 2000 EDUs from the heart and artery disease group) is the corpus for semi-automatic annotations of the wrdCo concepts (as the wrdCoc features) and the relation-class of each wrdCoc pair by the experts on the next step of Section 4.1.2 where this annotated corpus is used as a learning corpus in Section 4.2. The second part is a test corpus which consists of 2000 EDUs from the diabetes and kidney disease group and 2000 EDUs from the heart and artery disease group. The test corpus of each disease group is used for (1) determining and collecting wrdCoc pairs as CEpairs having CErel into WCP in Section 4.3 and (2) extracting the causal pathways in Section 4.4.

#### 4.1.2. Semi-Automatic Corpus Annotation

The semi-automatic corpus annotation of each disease group consists of the wrdCoc feature annotation on the wrdCo expressions and the CErel annotation on each wrdCo–expression pairs. We semi-automatically annotate the corpus by using an element of a discourse-marker cue set, {'ทำให้/*causing*', 'ดังนั้น/*because*', 'เพราะว่า/*since*'}, to anchor on the corpus documents for obtaining predicate verbs, $v_a$, ($v_a \in z \cup V_{inf}$ ; $a$ = 1, 2, …, *numofPredicateVerbs*) from all EDU occurrences right before and right after the anchored causal-cue set elements. Then we obtain a V-pair set = the result of the self-Cartesian

product (V × V). We use all V-pair set elements to search two adjacent predicate verbs ($v_{a1}$ $v_{a2}$ where $v_{a1}, v_{a2} \in$ V; $v_{a1} <> v_{a2}$; $a1 <> a2$) of two adjacent EDU occurrences along with automatically annotating the $v_a$, $w_{1,d}$, and $w_{2,e}$ terms of two adjacent EDUs' wrdCo expressions for the wrdCo concept annotation as the wrdCoc features by the experts selecting the concepts from Lexitron Dictionary after the Thai-to-English translation. Where the concepts from Lexitron Dictionary are referred to Thai Encyclopedia (https://www.saranukromthai.or.th/index2.php), MeSH (https://www.ncbi.nlm.nih.gov/mesh accessed on 10 August 2021), and Wordnet [28] (http://word-net.princeton.edu/obtain accessed on 10 August 2021). Additionally, the relation class (CErel/nonCErel) between two annotated wrdCoc features as a wrdCoc pair (or CEpair) on the annotated corpus is also annotated by the expert as shown in Figure 6 for learning the relation-class in Section 4.2. Both the wrdCo expressions and the wrdCoc features from both disease groups are collected into wrdCo-Concept Table (see Table 1) containing several wrdCo expressions with the same wrdCoc feature where the duplicate entries are eliminated.

**Figure 6.** Annotation of wrdCo concepts or wrdCoc features including CErel/nonCErel Class between wrdCoc pair where v, w1, and w2 symbols in a wrdCo tag is $v_a$, $w_{1,d}$, and $w_{2,e}$ terms/elements respectively of wrdCoPattern.

**Table 1.** wrdCo-Concept Table from the annotated corpus.

| wrdCo Expression Based on wrdCoPattern | | | wrdCoc Feature |
|---|---|---|---|
| V | W1 | W2 | |
| ÇÈÇí/*deposit* <deposit> | ž″íŝ/*fat* <fat> | Ëûŝñ¦Åº²ᴁo/*blood vessel wall* <wall> | **beDeposited(fat,bloodVessel)** |
| ²ₒᴁ/*deposit* <deposit> | ž″íŝ/*fat* <fat> | Çñ²ᴁ°ùº³/*artery* <bloodVessel> | **beDeposited(fat,bloodVessel)** |
| Eᴁ/*deposit* <deposit> | ₒᴁᴁ²íŝ/*having fat* <fat> | Ëûŝñ¦Åº²ᴁo/*blood vessel wall* <wall> | **beDeposited(fat,bloodVessel)** |
| íᴁ″íŝ/*have fat* <haveFat> | ñ¦Åº²ᴁ°ùº³/*artery* <bloodVessel> | ÇÈÇí/*deposit* <deposit> | **beDeposited(fat,bloodVessel)** |
| ²ₒᴁ/*deposit* <deposit> | ž″íŝ/*fat* <fat> | ôÈ₀ŝ/*plaque* <bePlaque> | **beDeposited(fat,bePlaque)** |
| ᴁôÈₒᴁž″íŝ/*is fatty-plaque* <bePlaque> | Çᴁᴁ̂ŝûñ¦Åº²ᴁº /*embolism* <embolism> | null | **bePlaque(embolism)** |
| ₒÅₒᴁ/*form* <form> | ôÈₒᴁž″íŝ/*fatty-plaque* <plaque> | ñûûᴁ/*thick* <thick> | **Form(plaque,thick)** |
| ôᴁû″ᴁ/*be narrow* <beNarrow> | ñ¦Åº²ᴁo/*blood vessel* <bloodVessel> | null | **beNarrow(bloodVessel)** |
| ôᴁû″ᴁ/*be narrow* <beNarrow> | ñ¦Åº²ᴁo/*blood vessel* <bloodVessel> | ¦³/*more* <more> | **beNarrow(bloodVessel)** |
| ñ¦Å²ᴁ/*supply* <supply> | ²ᴁo/*blood* <blood> | ₒᴁûôᴁᴁᴁ/*myocardium* < myocardium> | **beSuppliedInsufficiently (blood, myocardium)** |
| ″ô/*lack* <lack | ₒᴁûôᴁᴁᴁ /*myocardium* < myocardium> | ²ᴁo/*blood* <blood> | **beSuppliedInsufficiently (blood, myocardium)** |
| ž¬²ᴁ/*supply* <supply> | ²ᴁo/*blood* <blood> | ûₒᴁᴁ/*tissue* <tissue> | **beSuppliedInsufficiently (blood, tissue)** |
| ″ô/*lack* <lack | Çíŝᴁ/*brain* <brain> | ²ᴁo/*blood* <blood> | **beSuppliedInsufficiently (blood, brain)** |
| ²ₒᴁ/*occur* <occur> | ₒᴁûᴁÇₒᴁᴁᴁíÂ /*chemical forming* < chemicalForming> | null | **Occur(chemicalForming)** |
| ²ₒᴁ/*occur* <occur> | ñ¦Åº²ᴁo/*blood vessel* <bloodVessel> | ₒᴁšÇ₄/*inflammation* < inflammation> | **Occur (bloodVessel,inflammation)** |
| ²ₒᴁ″ô/*occur* <occur> | ₒᴁšÇ₄/*inflammation* < inflammation> | ñ¦Åº²ᴁo/*blood vessel* <bloodVessel> | **Occur (bloodVessel,inflammation)** |
| ù″ôᴁ/*be stiff* <beStiff> | ñ¦Åº²ᴁo/*blood vessel* <bloodVessel> | null | **beStiff(bloodVessel)** |
| Åᴁôŝ/*get clogged* <beClogged> | ñ¦Åº²ᴁo/*blood vessel* <bloodVessel> | null | **beClogged(bloodVessel)** |
| íᴁᴁº₀ᴁₒᴁûñ²ᴁÇᴁ /*have hyperglycaemia* <haveHyperglycaemia> | ËᴁÃᴁ/*patient* <person> | null | **haveHyperglycaemia(person)** |
| ùôᴁ/*break* <break> | ñ¦Åº²ᴁo/*blood vessel* <bloodVessel> | null | **beBroken(bloodVessel)** |
| ôᴁôᴁôᴁᴁ/*damage* <beDamaged> | ûₒᴁᴁ/*tissue* <tissue> | null | **beDamaged(tissue)** |
| ………………….. | ………………….. | ……………… | ………………….. |

## 4.2. CErel Learning on Each wrdCoc Pair

The objective of this step is CErel learning on each wrdCoc pair (which is a wrdCoc<sub>EDU</sub> pair as CEpair) with the CErel/nonCErel class from the annotated corpus used

as the learning corpus to obtain WCP of each disease group in the next section. Regarding the annotated corpus of each disease group from Section 4.1, each annotated corpus contains several EDUs with the wrdCoc-pair class annotations by the wrdCocPair tag. The wrdCoc features of each disease group, e.g., a CwrdCoc feature and a EwrdCoc feature (where CwrdCoc is a causative wrdCo concept; EwrdCoc is an effect wrdCo concept), are obtained by the wrdCo tag containing 'Concept' and 'type' (Figure 6). All annotated wrdCoc pairs as CwrdCoc, EwrdCoc pairs with CErel/nonCErel by the wrdCocPair tag of each disease group are used for learning CErel by NB, SVM, and LR based on ten-fold cross validation.

(a) NB [18]. The NB learning results of each disease group by this step based on using Weka (http://www.cs.wakato.ac.nz/ml/weak/ accessed on 10 August 2021) are the probabilities of CErel and nonCErel of CwrdCoc features and EwrdCoc features in wrdCoc pairs as shown in Table 2. Where CwrdCoc $\in$ CWC which is a causative-wrdCo-concept set; EwrdCoc $\in$ EWC which is an effect-wrdCo-concept set; and CWC$\cap$EWC$\neq\varnothing$.

**Table 2.** Show the CErel and nonCErel probabilities of CwrdCoc features and EwrdCoc features in wrdCoc pairs of the diabetes and kidney disease group and the heart and artery disease group.

| Disease Group | CwrdCoc | CErel | Noncerel | EwrdCoc | CErel | Noncerel |
|---|---|---|---|---|---|---|
| Diabetes & Kidney Disease Group | <beLost(protein,urine)> | 0.0698 | 0.0784 | <beLost(protein,urine)> | 0.0074 | 0.0066 |
| | <beFailure(kidney)> | 0.1581 | 0.0784 | <beFailure(kidney)> | 0.0355 | 0.0333 |
| | <haveHyperglycaemia (person)> | 0.0452 | 0.0294 | <haveHyperglycaemia (person)> | 0.0411 | 0.0200 |
| | <notTakeSugar(body)> | 0.0266 | 0.0392 | <notTakeSugar(body)> | 0.0112 | 0.0133 |
| | <beNarrow(bloodVessel)> | 0.0185 | 0.0098 | <beNarrow(bloodVessel)> | 0.0374 | 0.0533 |
| | …………… | … | … | ……………... | … | … |
| Heart & Artery Disease Group | <beDeposited(fat, bloodVessel)> | 0.0540 | 0.0288 | <beDeposited(fat, bloodVessel)> | 0.0011 | 0.0193 |
| | <beThick(bloodVessel)> | 0.0149 | 0.0288 | <beThick(bloodVessel)> | 0.0169 | 0.0038 |
| | <beDamages(bloodVesselWall)> | 0.0218 | 0.0041 | <beDamages(bloodVesselWall)> | 0.0101 | 0.0038 |
| | <becomeInflamed(bloodVessel)> | 0.1323 | 0.1028 | <becomeInflamed(bloodVessel)> | 0.0079 | 0.0115 |
| | <beClogged(bloodVessel,organ)> | 0.0448 | 0.0288 | <beClogged(bloodVessel,organ)> | 0.0418 | 0.0424 |
| | …………… | … | … | ……………... | … | … |

(b) SVM [19]. The SVM learning is a linear binary classification applied to classify the CErel and nonCErel of each wrdCoc pairs from the annotated corpus by using Weka. This linear function, $f(\mathbf{x})$, of the input $\mathbf{x} = (x_1, x_2, …, x_n)$ assigned to the Cerel class if $f(x) > 0$, and otherwise to the nonCErel class, is as Equation (3).

$$f(x) = \langle w \cdot x \rangle + b$$
$$= \sum_{j=1}^{n} w_j x_j + b \tag{3}$$

where $x$ is a dichotomous vector number, w is weight vector, $b$ is bias, and $(w, b) \in \mathbb{R}^n \times \mathbb{R}$ are the parameters that control the function. The SVM learning is to determine $w_j$ and $b$ for each wrdCoc feature ($x_j$) which is either a CwrdCoc feature or a EwrdCoc feature in each wrdCoc pair with CErel or nonCErel from the annotated corpus of each disease-group.

(c) LR [20]. The logistic regression model of the research is based on the linear logistic regression with binary vector data. The distinguishing feature of the logistic regression model is that the variable is binary or dichotomous. Usually, the input data with any value from negative to positive infinity would be used to establish which attributions are influential in predicting the given outcome with values between 0 and 1, and hence is interpretable as a probability. The logistic function can be written as:



$$F(x) = \frac{1}{1 + e^{-(\beta_0 + \beta_1 x_1 + \beta_2 x_2)}} \tag{4}$$

$F(x)$ is interpreted as the probability of the given outcome to be predicted where $x_1$ and $x_2$ are attribute variables; $\beta_0$ is bias; and $\beta_1$, and $\beta_2$ are the model estimators which play the role of momentum for each attribute. The LR learning is to determine $\beta_0$, $\beta_1$, and $\beta_2$ for each CwrdCoc feature and each EwrdCoc feature as $x_1$ and $x_2$ features respectively in each wrdCoc pair (CwrdCoc, EwrdCoc) with either the positive/CErel class or the negative/nonCErel class formed by supervised learning on the learning corpus of each disease-group.

The learning results by NB, SVM, and LR models are the estimators which are used for determining wrdCoc pairs having CErel from the test corpus of each disease group in the next step of Section 4.3. Moreover, all precisions of learning by NB, SVM, and LR from the learning corpus of each disease group are greater than 0.8.

### 4.3. Determination of wrdCoc Pairs Having CErel

The WC elements are determined from all wrdCo expressions on the test corpus of each disease group by the wrdCo-expression matching between the wrdCo expressions on this test corpus and the wrdCo expressions on wrdCo-Concept Table (Table1) to obtain the wrdCoc features or the WC elements. The result of the self-Cartesian product (WC × WC) is a wrdCo-concept ordered pair set which is used for determining and collecting the wrdCoc pairs having CErel into WCP of each disease group by the following NB, SVM, and LR.

(a) NB. Regarding Equation (5) and the CErel and nonCErel probabilities of CwrdCoc and EwrdCoc (Table 2), the CwrdCoc EwrdCoc pairs as the wrdCoc pairs having Cause-EffectRelationClass as CErel is determined from the self-Cartesian product (WC × WC) result and then collected into WCP of each disease group on which CwrdCoc and EwrdCoc are independent.

$$\begin{aligned}
\text{Cause-EffectRelationClass} &= \underset{class \in Class}{\text{argmax }} P(class | CwrdCoc, EwrdCoc) \\
&= \underset{class \in Class}{\text{argmax }} P(CwrdCoc | class)P(EwrdCoc | class)P(class)
\end{aligned}$$

where $CwrdCoc$ is a causative-wrdCo concept; $\qquad\qquad\qquad$ (5)

$\qquad EwrdCoc$ is an effect-wrdCo concept;

$\qquad$ Class = {"CErel", "nonCErel"}.

(b) SVM. The bias, $b$, and the weight vector, w, of the CWC elements and the EWC elements in the wrdCoc pairs from the SVM learning (Section 4.2 (b)) are used to determine and collect the CwrdCoc EwrdCoc pairs as the wrdCoc pairs having the CErel class into WCP of each disease group from the self-Cartesian product (WC × WC) result with Equation (3).

(c) LR the research applies Equation (4) along with Equation (6) to determine the CErel class between the CWC elements and the EWC elements in the wrdCoc pairs from both the positive/CErel class determination and the negative/nonCErel class determination by using the estimators from the LR learning (Section 4.2 (c)).

$$\text{CErelClass} = \text{Max}(F(x)_{\text{CErelClass}}, F(x)_{\text{nonCErelClass}}) \tag{6}$$

According to (6), $x_1$ and $x_2$ as CwrdCoc and $x_2$ as EwrdCoc are the attribute variable pair of each wrdCoc pair from the test corpus of each disease group where $\beta_0$, $\beta_1$, and $\beta_2$ of CwrdCoc and EwrdCoc are obtained by the supervised learning with LR on the learning corpus of each disease group. The wrdCoc pair (or the CwrdCoc and EwrdCoc pair) with the CErel class is determined and collected into WCP of each disease group from the self-Cartesian product (WC × WC) result.

### 4.4. Causal Pathway Extraction

All wrdCoc features per the test-corpus document of each disease group are sequentially collected in an array of wrdCoc features ($wcc$ [ ]) after the wrdCo-expression matching between the wrdCo expressions of this test-corpus document and the wrdCo expressions on wrdCo-Concept Table (Table1) to obtain the wrdCoc features. The causal pathways are then extracted by the wrdCoc-pair matching between $wcp_k$ ($wcp_k \in$ WCP; $k$ = 1, 2, …, *numberOfWCPelements*) and each wrdCoc pair in $wcc$ [ ] as a $wcc_{[ct]}$ $wcc_{[ct+1]}$ pair ($ct$ = 1, 2, …, *numberOFwrdCocFromTestCorpusDocument*) by sliding a window size of two consecutive wrdCoc features ($wcc_{[ct]}$ $wcc_{[ct+1]}$) with one wrdCoc distance ($wcc_{[ct++]}$) on $wcc$ [ ]. We stop sliding the window if there is no match on wrdCoc-pair matching. We then obtain a causal pathway as shown in Algorithm 1 where CEpair$_i$ is a wrdCoc pair ($wcc_{[ct]}$ $wcc_{[ct+1]}$) in $wcc$ [ ]; and allPathways (which is an array of arrayList with an 'ɑ variable of an array size) contains several causal pathways.

---

**Algorithm 1** Causal Pathway Extraction

CAUSAL_PATHWAY_EXTRACTION
/*  (Extraction of several CEpairi sequences as causal pathways.)
/*  Assume that each EDU is represented by (NP1   VP).
/*  L is a list of EDUs from one test-corpus document after stemming words and the stop word removal.
/*  CEpairi is a wrdCoc pair with index i of the causal pathway.
/*  wcc[ ] is an array of wrdCoc and is collected from this test corpus.
/*  WCP is a set of wrdCoc pairs having CErel.
1: **ct = 1;j = 1;ct = 1; a = 0 ;   string wcc[];**
2: **ArrayList<string> []allPathways = new ArrayList[a];**
/*   **array of arrayList.**
3: **while j≤ Length[L] do**
4: **{₁ wcexpⱼ = getWrdCo(EDUⱼ);**
/* Get wrdCo expression of EDUj from the test corpus.
5:    **If   wcexpⱼ.v∈Vstrong ∪ Vinf then**
/* **wcexpⱼ.v is a predicate verb   vₐ   on a wrdCo expression**
**with index j.**
6:     **{   wcc_{[ct]}= getWrdCoConcept ; ct++};**
/* **getWrdCoConcept   by the wrdCo-expression matching between wrdCo**
**expressions of the test-corpus document and the wrdCo expressions**
**with the wrdCoc features on wrdCo-Concept Table (Table1).**
7:   **j++ }1 ;**
8: **count = ct; ct = 1; i = 1; flagce = 0; flagec = 0; fl = 0;**
9: **while ct ≤ count-1 do**
10: **{₁ while (wcc_{[ct]} + wcc_{[ct+1]}∈ WCP)   ∧   (ct ≤ count-1)   do**
/* **a causal pathway extraction** by wrdCoc-pair matching
wcc_{[ct]}+wcc_{[ct+1]}) among wcp_k (where wcp_k∈WCP).
11:     **{₂ CEpairᵢ = wcc_{[ct]}+ wcc_{[ct+1]} ;**
/* CErel occurs on Text as EDUCauseEDUEffect.
12:       **If flagce = 0 then {a++; flagce = 1};**
13:       **allPathways_{[a]}.AddNewCause¬EffectPair(CEpairᵢ); i++;ct++ }₂;**
14:     **If flagce = 1 then { flagce = 0; fl = 1; i = 1};**
15:       **while (wcc_{[ct+1]} + wcc_{[ct]} ∈  wcpk) ∧ (ct ≤  count-1) do**
/* **another causal pathway extraction** by wrdCoc-pair
matching (wcc_{[ct+1]}+wcc_{[ct]})among wcp_k (where wcp_k∈WCP).
16:         **{₃ CEpairᵢ = wcc_{[ct+1]} + wcc_{[ct]} ;**

/* CErel occurs on Text as $EDU_{Effect}EDU_{Cause}$.

/* $wcc_{[ct+1]}$ is Cause and $wcc_{[ct]}$ is Effect.

17:         **If flagec = 0 then {a++; flagec = 1};**

18:       **allPathways[a].AddNewCause¬EffectConceptPair(CEpair*i*);**

19:        **i++; ct++}₃ ;**

20:     **If flagec = 1 then { flagec = 0; fl = 1 ; I = 1};**

21:    **If fl = 0 then ct++;**

22:     **else fl = 0;**    **}₁**    ;

23: **}Return   allPathways**    **/\*** Return causal pathways.

*4.5. Implicit-Mediator Indication and Representation with Explicit-Mediators*

The correct extracted causal pathways of the allPathways result for each disease group by the CausalPathwayExtraction algorithm on Section 4.4 consists of the explicit mediator causal pathways and the implicit mediator causal pathways. The allPathways result of each disease group also contains some duplicate causal pathways. Therefore, allPathways is sorted and then is eliminated the duplicate causal pathways to become PathWays (which is an array of arrayList with an updated 'ɗ' variable) before indicating the implicit mediators on the correct extracted causal pathways. With regard to PathWays of each disease group, the causal pathways containing the explicit mediators represented by EDUs' wrdCoc features are collected into the dynamic template as the ExplicitCEpairWithCErelPathways template (see Section 3.3) which is an ExplicitPath variable in an ExplicitCausalPathwayRepresentation algorithm (Algorithm 2) whilst the causal pathways containing the implicit mediators are collected into an ImplicitPath variable as a temporary template.

The implicit mediator indication and representation on PathWays with the explicit mediators from ExplicitPath/the ExplicitCEpairWithCErelPathways template is based on the following steps of the ExplicitCausalPathwayRepresentation algorithm (Algorithm 2):

(1) Determine TransCEPair from all correct extracted causal pathways ($Pathways_{[\alpha]}$ ; $\alpha$ = 1, 2, …, ɗ).

(2) Use TransCEPair to indicate $CEpair_i$ having the implicit mediator as the causal transitivity on $Pathways_{[\alpha]}$; if $Pathways_{[\alpha]}$ contains $CEpair_i \notin$ TransCEPair (where $i$ = 1, 2, …, *numOfCauseEffectConceptPairs*), $Pathways_{[\alpha]}$ is the explicit-mediator causal pathway; and is added to $ExplicitPath_{[\alpha]}$.

(3) If $Pathways_{[\alpha]}.CEpair_i \in$ TransCEPair, we mark '\*'on $CEpair_i$ having causal transitivity or the implicit mediator, add $*CEpair_i$ and subsequent $CEpair_{i+1}$, $CEpair_{i+2}$, …, $CEpair_{numOfCauseEffectConceptPairs}$ of $Pathways_{[\alpha]}$ to $ImplicitjPath_{[\alpha]}$, and add $CEpair_1, CEpair_2…CEpair_{i-1}$ of $Pathways_{[\alpha]}$ to $ExplicitPath_{[\alpha]}$.

(4) Replace $*CEpair_i$ with the explicit mediator(s) of $CEpair_{idd}$ from ExplicitPath as shown in Algorithm 2.

---

**Algorithm 2** Explicit Causal Pathway Representation

**EXPLICIT_CAUSAL_PATHWAY_REPRESENTATION (ArrayList\<string\> []Pathways;** a)

/*    Assume Pathways is allPathways (byFig.7) with eliminating the duplicate causal pathways.

/*    trsvSet is TransCEPair which is a set of CEpairs with CErel to be transitive.

/*   ExplicitPath is a dynamic ExplicitCEpairWithCErelPathways template.

1: trsv = 0; check1 = 0; trsvSet←∅;

2: ArrayList\<string\> []ExplicitPath = new ArrayList[a];

ArrayList\<string\> []ImplicitPath = new ArrayList[a];

ArrayList\<String\> fill = new ArrayList\<\>();

3: num1 = a

/* **where a is the number of Pathways elements(the Pathways array size).**

4: For ($\alpha$ = 1 to num1 ; $\alpha$++) /***determine trsvSet** from Transitive Closure

5: {trsvSet$\leftarrow$trsvSet $\cup$ Pathways[$\alpha$].transitiveClosureDetermination };

6: For ($\alpha$ = 1 to num1 ; $\alpha$++)

7: {$_1$ i = 1 ; /***collect Explicit-Mediator Causal Pathways to ExplicitPath.**

8:        while (i $\leqslant$ Pathways[$\alpha$].numberOfCauseEffectConceptPairs) $\wedge$ (Pathways[$\alpha$].Get(CEpair$_i$)$\notin$TrsvSet)    do

/***add explicitCEpair$_i$ to ExplicitPath.**

9:          { ExplicitPath[$\alpha$].Add(Pathways[$\alpha$].Get(CEpair$_i$)) ; i++    }

10:        while (i $\leqslant$ Pathways[$\alpha$].numberOfCauseEffectConceptPairs) do

11:      {If (Pathways[$\alpha$].Get(CEpair$_i$)$\in$TrsvSet)      then

/* **Identify and mark CEpairi having implicit Mediator as**

**causalTransitivity with '*'; and then**

**add '*'CEpair$_i$&all subsequent CEpair$_i$,...to ImplicitPath.**

12:              ImplicitPath[$\alpha$].Add('*'+Pathways[$\alpha$].Get(CEpairi))

else ImplicitPath[$\alpha$].Add(Pathways[$\alpha$].Get(CEpairi)); i++ } }1

/* **replace'*'CEpairi on ImplicitPath with explicit mediator from**

**ExplicitPath**.

13:    while      check1 = 0 do

14:    {$_1$For ($\alpha$ =1 to num1 ; $\alpha$++)

15:      {$_2$If    ImplicitPath[$\alpha$]isNotEmpty    then

16:        {$_3$idi = 1 ; id = 1;    check2 = 0;    fill.clear()

17:            while idi $\leqslant$ImplicitPath[$\alpha$].numberOfCEpairs do

/* **find'*'CEpairidion ImplicitPath.**

18:          {$_4$ If ImplicitPath[$\alpha$].Get(CEpairidi).contain('*') = true then

19:              {$_5$ ImplicitPath[$\alpha$].Get(CEpairidi).replace('*','');

20:                  C = CEpair$_{idi}$.wcc$_{idi}$;    E = CEpair$_{idi}$.wcc$_{idi+1}$;

/* **get C/cause & E/effect.**

b = 1; idd = 1; f1 = 0; f2 = 0; f3 = 0;

21:                    while b $\leqslant$  num1 $\wedge$ f1 = 0 do

/***find an explicit mediator: C+ mediator(s)(CEpairidi...)+E**

**from ExplicitPath.**

22:    {$_6$ while idd $\leqslant$ExplicitPath[b].numberOfCauseEffectConceptPairs $\wedge$f3=0 do

23:        {$_7$ If C = ExplicitPath[b].Get(CEpairidd).wccidd then    f2 = 1

24:            elseIf(E=ExplicitPath[b].Get(CEpair$_{idd}$).wcc$_{idd+1}$)$\wedge$f2 = 1 then f3=1;

25:            If f2 = 1$\vee$f3 = 1 then {fill$_{[id]}$=ExplicitPath[b].Get(CEpair$_{idd}$); id++};

26:            idd++ }$_7$ ;    /* **fill contains C+ mediator(s)(CEpair$_{idi}$...)+E.**

27:      If f3 = 0 then   {idd = 1; f2 = 0; fill.clear()}   /* **no E in fill.**

28:        else {f1 = 1; ExplicitPath[$\alpha$].addAll(fill); check2 = 1};

/* **add fill to ExplicitPath.**

Id = 1; b++ }$_6$   }$_5$

29:        elseIf$_5$ ImplicitPath[$\alpha$].Get(CEpairidi).contain('*') = false $\wedge$ check2 = 1 then

{ExplicitPath[$\alpha$].add(ImplicitPath[$\alpha$].Get(CEpair$_{idi}$)}$_5$

idi++ }$_4$

30:    If check2 = 1 then {ImplicitPath[$\alpha$].clear(); check2 = 0 }   }$_3$   }$_2$

31: For ($\alpha$ = 1 to num1 ; $\alpha$++ )   /* **check ImplicitPath being empty.**

32:    {If    ImplicitPath[$\alpha$] isNotEmpty then check2 = 1 };

33: If check2 = 0 then check1 = 1;
34: }₁; ExplicitPath.sortRowOfArrayOfArrayList;
ExplicitPath.removeDuplicateRow;
35: }Return **ExplicitPath**

(5)  Check the ExplicitPath result from the ExplicitCausalPathwayRepresentation algorithm does not contain the implicit-mediator(s)/the causal transitivity by comparing trsvSet (line no. 5 of Algorithm 2) to TransCEPair determined from the ExplicitPath result on line no. 35 of Algorithm 2. If the TransCEPair determination from ExplicitPath is the same as trsvSet, ExplicitPath contains the explicit- mediator causal pathways without the causal transitivity or the implicit mediator, otherwise the ExplicitCausalPathwayRepresentation algorithm is re-executed after copying ExplicitPath to Pathways of Algorithm 2 and then setting ExplicitPath to empty.

Therefore, the representation of the correct extracted causal pathways with the explicit mediators by Algorithm 2 is shown in Figure 7.

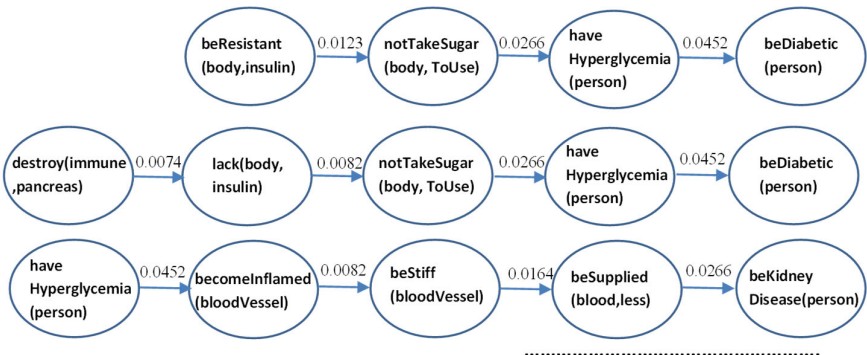

**Figure 7.** The representation of the correct extracted causal pathways with the explicit mediators of the diabetes and kidney disease group where the numeric label on each arrow represents the CErel probability of each CEpair by NB.

## 5. Evaluation and Discussion

The test corpus of 4000 EDUs employed to evaluate the proposed methodology for the causal pathway extraction through determining wrdCoc pairs having CErel is collected from the downloaded disease documents on Thai hospital web-boards. The test corpus consists of 2000 EDUs from the diabetes and kidney disease group documents and the 2000 EDUs from the heart and artery disease group documents. There are three evaluations, 1) the determination of wrdCoc pairs having CErel, 2) the causal pathway extraction, and 3) the implicit-mediator indication and representation with the explicit mediators from the dynamic template.

### 5.1. Determination of wrdCoc Pairs Having CErel

The evaluation results of extracting the EDU-concept pairs/wrdCoc pairs having CErel from the documents of the diabetes and kidney disease group and the heart and artery disease group are the precisions and the recalls based on three experts with max win voting as shown in Table 3 including the number of different wrdCoc features which results in the frequencies of wrdCoc features as shown in Figure 8.

**Table 3.** The accuracy of determining wrdCoc pairs having CErel.

| Disease Group (2000 EDUs/Group) | #of Differrent wrdCoc Features | | Extraction of wrdCoc Pairs Having CErel | | | | | |
|---|---|---|---|---|---|---|---|---|
| | | | NB | | SVM | | LR | |
| | Cause | Effect | Precision | Recall | Precision | Recall | Precision | Recall |
| Diabetes and kidney disease group | 40 | 92 | 0.844 | 0.777 | **0.893** | **0.803** | 0.877 | 0.795 |
| Heart and artery disease group | 93 | 110 | 0.826 | 0.746 | **0.841** | **0.760** | 0.831 | 0.754 |

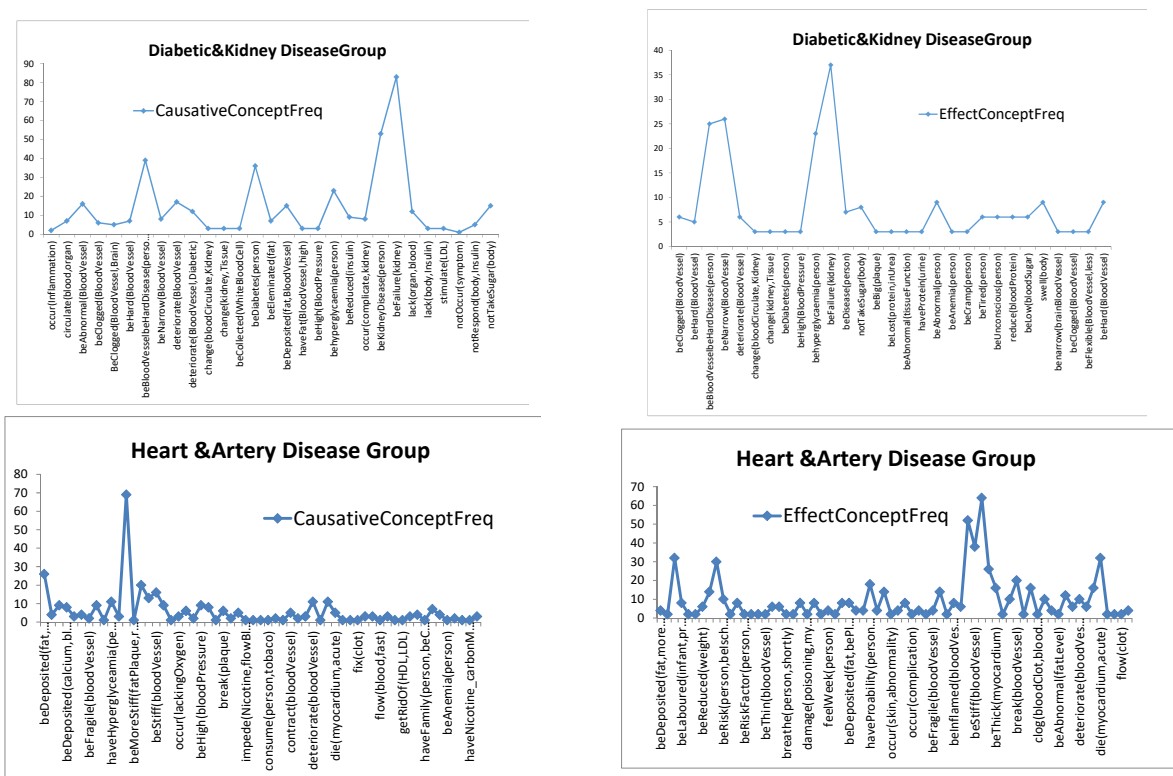

**Figure 8.** Show the wrdCoc frequencies with the causative concepts and the effect concepts from the diabetes and kidney disease group and the heart and artery disease group.

From Table 3, the average precisions of extracting wrdCoc pairs having CErel from the documents of the diabetes and kidney disease group and the heart and artery disease group are 0.871 and 0.833 respectively, with the average recalls of 0.791 and 0.753 and the average F-score of 0.830 and 0.791 respectively. Whereas the causality or cause-effect relation extraction from the previous research [7] based on the probabilities of words on NP pair and the cue phrase probability from the complex sentence or two simple sentences from the medical domain has an F-score of 0.774. With regard to our research results on Table 3, the reason for the diabetes and kidney disease group having the higher precision and recall of extracting wrdCoc pairs having CErel than the heart and artery disease group is that the heart and artery disease group have more diversity of the wrdCoc features in both the causative concepts and the effect concepts than the diabetes and kidney disease group. The high diversity of wrdCoc features results in low frequencies of most wrdCoc features. In addition, there are some dependency occurrences among wrdCoc features, e.g.,

EDU1: 'haveHyperglycaemia(person)' → EDU2:'becomeInflamed(BloodVessel)', where EDU1's wrdCoc and EDU2's wrdCoc mostly occur as a cause-effect relation or a dependency occurrence on documents but some documents contain EDU1's wrdCoc followed by EDU2's wrdCoc without the cause-effect relation. Thus, the wrdCo diversity and the wrdCoc dependency result in SVM having highest precision in both the diabetes and kidney disease group and the heart and artery disease group. However, both the diabetes and kidney disease group and the heart and artery disease group have low recalls because the diversity of wrdCo expressions occurs on the downloaded documents of both disease groups.

### 5.2. Causal Pathway Extraction

The causal pathway extraction from the test corpus is evaluated by the precision and recall based on three experts with max wins voting as shown in Table 4.

**Table 4.** The accuracy of extracting causal pathways.

| Disease Group (2000 EDUs/Group) | Causal Pathway Extraction | |
|---|---|---|
| | Precision | Recall |
| Diabetes and Kidney Disease Group | 0.840 | 0.724 |
| Heart and artery Disease Group | 0.828 | 0.706 |

The causal pathways determination from the documents of two disease group as shown in Table 4 have an average precision of 0.834 with the average recall of 0.715. The reason for having low recall of extracting causal pathways from the documents is that some causal pathways start with EDUs containing the causative/effect concept expressed by either NP1 or NP2 as shown in the following EDU1 of Example 5 instead of the predicate verb or Verb on the general linguistic expression in Figure 1.

**Example 5.**

*EDU1.* "ꪺꪹꪻ꪿ꫀꪹꪶꪹꫂ꫁ꪶ꪿ꪹꪶꪺꪶ꫁ꪶ/*
*Kidney disease is caused by being diabetic for a long time".*
*(ꪺꪹꪻ꪿ꫀ/Kidney disease)/NP1*
*(ꪹꪶꪻ/is caused by ( ꪶꫂꪶ꫁ꪹꪶꪺꪶ꫁ꪶꪺꪶ/being diabetic for a long time)/NP2)/VP.*

*EDU2.* "ꪺꪻꪹꪹꪶ꫁ꪹꪹ꪿ꪻꪹꪺ꫁ꪶ/*Causing artery wall to be destroyed".*
*(ꪺꪻꪹ/Causing)/conj (ꪹꪹꪶ꫁ꪹꪹ꪿ꪻ/artery wall)/NP2*
*((ꪺꪺꪶ꫁ꪶ/is destroyed)/Verb_strong)/VP*

*EDU3.* "ꪺ꫁ꪶꪻꪺꪹꪹꪶꪹ꫁ꪶꪹꪹ꪿ꪹꪹꪺꪶꪹꪶꪹꪺꪶ/*Then the filtration function of the kidneys will deteriorate".*
ꪺ꫁ꪶ/*Then ( ꪺꪹꪹꪶꪹ꫁ꪶꪹꪹ꪿ꪹꪹꪺꪶ/the filtration function of the kidneys)/NP1*
*((ꪹꪹꪺꪶ/will deteriorate)/Verb_strong)/VP.*

*EDU4.* "ꪺꪻꪹꪹꪶꪹ꫁ꪺꪺꪶꪹꪶꪹꪹꪶꪹꪺꪶꪹ/*Causing protein to leak out in the urine".*
*(ꪺꪻꪹ/Causing)/conj (ꪹꪶꪹꪺꪶ/protein)/NP1*
*((ꪺꪺꪶꪹꪶ/leaks out)/Verb_strong ꪶꪹ/in (ꪹꪺꪶꪹꪹꪶꪹ/the urine)/NP2)/VP.*

However, the evaluation of the previous work [10] on extracting and constructing the causal chain/pathways from a large corpus on an on-line social media (tweets, news articles, and blogs) relied on time series through prediction of noun phrases as the next effect is 57% accuracy based on expert judgments whereas our causal pathways relied on the actual events/states with the causative/effect concepts.

### 5.3. Implicit-Mediator Indication and Representation with Explicit-Mediators

We evaluate the implicit mediator indication and representation with the explicit mediators (from the dynamic template) in term of a Likert scale (1 to 5) for concise and comprehensible representations of the correct extracted causal pathways. The evaluation results with the average scores (based on the Likert scale) of the concise and

comprehensible representations of Doc (which is the causal pathway representation by explanation on the documents) and Graph (which is the causal pathway representation by the correct extracted causal pathway with the explicit mediators from the documents) by the 30 end-users (who are non-professional persons) are presented on Table 5 and Figure 9 of both disease groups.

**Table 5.** The evaluation of the concise and comprehensible representations of Doc and Graph is based on scoring with the Likert scale (1 to 5).

| Disease Group | Concise Representation | | Comprehensible Representation | |
|---|---|---|---|---|
| | **Average Score by Doc Representation** | **Average Score by Graph Representation** | **Average Score by Doc Representation** | **Average Score by Graph Representation** |
| Diabetes and Kidney | 3 | 4.5 | 3 | 4.3 |
| Heart and Artery | 2.2 | 4.3 | 2.7 | 4.2 |

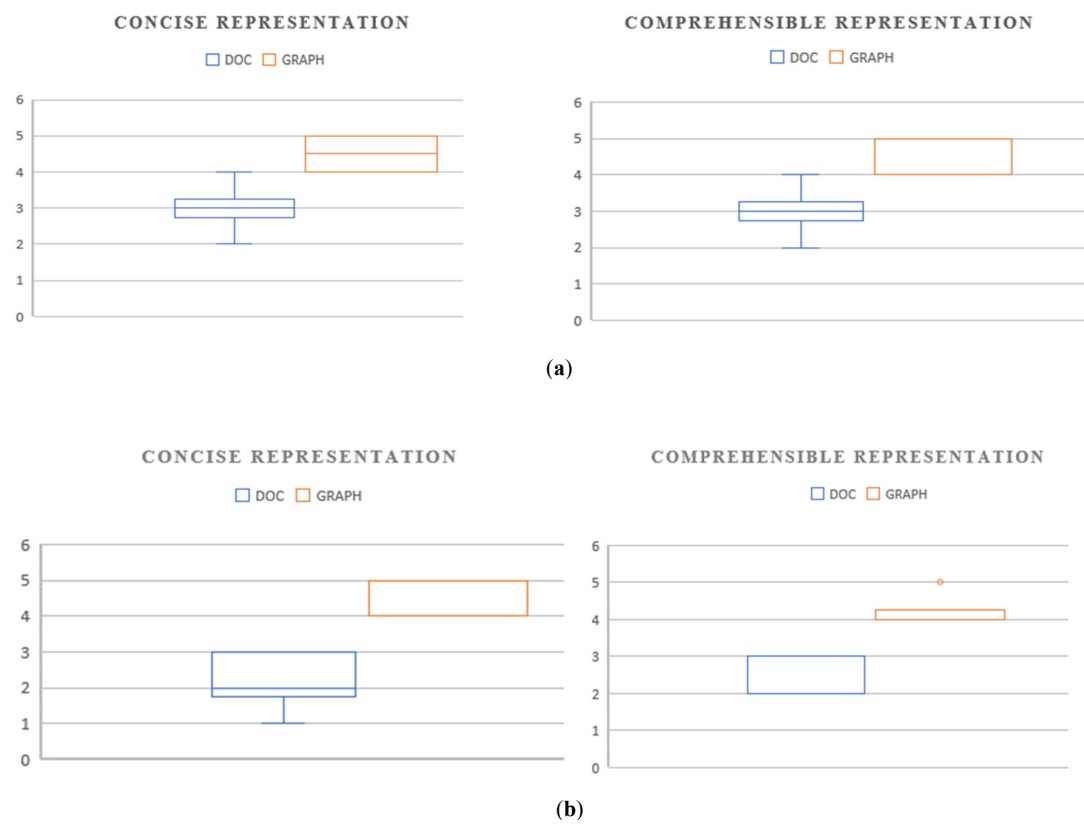

**Figure 9.** Show Box plot of the concise and comprehensible representations by Doc and Graph with the Likert scale 1–5. (**a**) Diabetes and Kidney Disease Group. (**b**) Heart and Artery Disease Group.

From Table 5 and Figure 9, Graph Representations of both disease groups have higher concise and higher comprehensible representations than Doc Representations of both disease groups. Moreover, from Table 5, the average scores of the concise representation and the comprehensible representation by Graph Representations from both disease groups are 4.4 and 4.25 respectively whereas the evaluation of the implicit knowledge completion on the event chain of actions from the Japanese web-blog corpus of the previous work [13] based on the similarity scores of event pairs on the chain with

the higher than thresholds is 3.0 (based on the Likert scale 1–5) without the CErel consideration between event-pairs on the chain.

According to the evaluation of the comprehensible representation of our research on both disease group, there are a few causal pathways requiring more explicit mediators for more clear representation as shown in Example 6.

**Example 6.** *A causal pathway representation of arteriosclerosis from the downloaded 'Heart Disease and Blood Vessel' documents on the hospital web-board:*

    *bedeposited(cholesterol,BloodVessel)→beInflamed(bloodVesselWall)→occur(sclerosis)*

*where "beInflamed(bloodVesselWall)" is an explicit mediator. Whilst the extracted causal pathway of the "Heart Disease and Blood Vessel" document from Thai Encyclopedia contain several explicit mediators as shown in the following*

    *beDeposited(cholesterol,BloodVessel)→oxidize(cholesterol,OxygenDirivative)→*
    *beInflame(bloodVesselWall)→consume(whiteBloodCell,fatParticle)→*
    *accumulate(whiteBloodCell, bloodVesselWall)→*
    *(beThickas(bloodVesselWall,plaque)*

*where (beThickas(vascularWall,plaque) is occur(sclerosis)*.

## 6. Conclusions

In this paper, we presented the extraction of the causal pathways containing the explicit and/or implicit mediators through learning and determining the wrdCoc pairs or CEpairs having CErel from the downloaded documents of the diabetes and kidney disease group and the heart and artery disease group on the Thai hospital web-boards. Where each explicit mediator is expressed on the document by an effect/causative-concept EDU represented by an EDU's wrdCoc feature. We also represent the implicit mediators by the explicit mediators within the correct extracted causal pathways. With regard to the limited literation of the causal pathway extraction from texts, the extracted causal pathways including the explicit mediator representation of our research supports the preliminary causal inference and also makes non-professionals understand an etiological pathway including disease complication through the social media for the compliance to the preventive treatments. Our proposed method of extracting and representing the causal pathways in terms of the explicit mediators even the implicit mediator occurrences from the documents is based on (1) the wrdCoc-pair matching between the wrdCoc pairs on the test corpus and the WCP elements through the sliding window on the test corpus for the causal pathway extraction where each wrdCoc feature is obtained by the wrdCo-expression matching between the wrdCo expressions on the test corpus and the wrdCo expressions on the wrdCo-Concept Table. In addition, the wrdCoc features from the wrdCo-expression matching are based on the $v_a$, $w_{1,d}$, and $w_{2,e}$ terms with complete matching as in [29]. Since the precisions of determining wrdCoc pairs having CErel from the learning corpus and the test corpus are consistent, the causal pathways extracted by the wrdCoc-pair matching are strengthened (where the WCP elements obtained by the correct determination of wrdCoc pairs having Cerel) And (2) applying the transitive closure to obtain TransCEPair for indicating the implicit mediators on the correct extracted causal pathways to represent these implicit mediators with the explicit ones from the dynamic template. To evaluate the proposed method, the accuracy of determining the wrdCoc pairs having CErel depends on both the diversity of the wrdCoc features (including the diversity of wrdCo expressions) and the dependency between wrdCoc features; which later affect to the causal pathway extraction and representation. In contrast to the previous researches, our proposed method provides three contributions: 1) the CErel or cause-effect relation determination with high F-scores of our research is based on a wrdCoc pair for representing an event concept pair expressed by two EDU's verb phrases with the NP1 head noun consideration whereas the cause-effect relation determinations of the previous researches are based on either the NP1-NP2 pairs [5–7,9–12] within one/two simple sentences or the Verb pairs within two EDUs [8]. The event/state occurrences with the causative/effect concepts on our corpus contain the verb

phrases expressions (which relate to the NP1s' head nouns) more than only the noun phrase expressions in the literature. 2) the causal pathway extraction with high precisions of our research is based on the actual event/state occurrences with the causative/effect concepts and also emphasizes on the boundary of the sequent wrdCoc pairs through the wrdCoc-pair matching between a wrdCoc pair of each slided-window on the test corpus and the WCP elements. Whereas the causal pathway/chain of the previous works are relied on either the prediction of the next effect from the previous noun phrase events based on the time series [10] or two steps of the cause-effect relation based on noun terms/phrases connected by either the similarity score [11] or the edge weights [12], e.g., 'A causes B' and 'B causes C' are connected by B similarity score without considering a boundary of a sequence event pairs having CErel. (3) our research applies the transitive closure and the dynamic template to indicate and represent the implicit mediator with the explicit mediators respectively to the correct extracted causal pathways with the high concise and clear comprehensible representations whereas the previous work [13]on the implicit knowledge completion of the event chains is based on the similarity scores between each event pairs from the corpus without the CErel consideration whilst our method of using the transitive closure and the dynamic template can apply to present the implicit knowledge completion of [13].

In the future, the temporal feature and the condition feature should be considered to increase the accuracy of the causal pathway extraction by reducing the wrdCoc diversity in terms of conditional cases. Moreover, the proposed method can also be applied in other languages, and the causal pathway extraction (Figure 7) can provide health literacy for non-professional persons to have clear comprehension of disease complications in order to follow the preventive treatments suggested by the physician.

**Author Contributions:** The following is a list of contributions by each author: Conceptualization, C.P.; methodology, C.P.; software, C.P.; validation, R.P.; formal analysis, R.P.; investigation, C. Pechsiri; resources, C.P.; data curation, C.P.; writing—original draft preparation, C.P.; writing—review and editing, C.P. and R.P.; visualization, C.P. and R.P.; supervision, C.P.; project administration, C.P.; funding acquisition, C.P. All authors have read and agreed to the published version of the manuscript.

**Funding:** This research received no external funding.

**Institutional Review Board Statement:** Not applicable.

**Informed Consent Statement:** Informed consent was obtained from all subjects involved in the study.

**Data Availability Statement:** The data for corpus preparation was obtained from hospitals website http://haamor.com/;http://www.bangkok health.com; http://www.si.mahidol.ac.th/sidoctor/e-pl/; https://www.bumrungrad.com; etc. accessed August 2021.

**Conflicts of Interest:** The authors declare no conflict of interest.

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
