# Peer review of "Causal Pathway Extraction from Web-Board Documents"

_applsci, doi:10.3390/app112110342_

Round 1

Reviewer 1 Report

In this paper, the authors propose a framework to extract the casual pathways through the casual relation determination especially for the Thai documents where the zero anaphora or the implicit noun phrase characteristics bring several problems. By using machine learning mechanism and linguistic phenomena, the proposed method determines CErel on each EDU-concept pair from the documents, extracts casual pathways from several EDU-concept pairs, and indicates the implicit mediators on the correct extracted causal pathways from the documents. Experimental results on documents show its high accurate extraction performance. However, I still have some comments for authors:

  1. Novelty and importance of target problem should be further illustrated that it has never been considered and addressed before.
  2. The authors descript that the proposed framework has advantages in at least three different parts, but there are no more detailed information about that compared with other methods.
  3. The author introduces a lot of examples to depict the process of the framework. But the detailed parts of examples are not classified in a proper way, maybe the author can provide a better way to describe.
  4. CEpairs has five pathways that are grouped into two kinds of different ways to express the example 5, and four pathways for example 6. How to determine the number of the pathways? And it is necessary for all the pathways?
  5. There are five steps in the proposed framework overviewed in figure 5, but the order and the novelty cannot be seen in it. Moreover, the description of figure 4 is not shown up completely that needs to place in a normal way. A same situation also emerges in table 1.
  6. The author can read more recent studies regarding feature representation, sparse data analysis, and other related feature extraction techniques to improve the method of this paper. For instance, the following papers are highly relevant:
  • a) A deep latent factor model for high-dimensional and sparse matrices in recommender systems, IEEE Transactions on Systems, Man, and Cybernetics: Systems 26, 2019.
  • b) Graph regularized Lp smooth non-negative matrix factorization for data representation, IEEE/CAA J. Autom. Sinica, vol. 6, no. 2, pp. 584-595, Mar. 2019.
  • c) A parallel matrix factorization based recommender by alternating stochastic gradient decent, Engineering Applications of Artificial Intelligence 25 (7), 1403-1412 37, 2012.
  • d) Regularizing deep neural networks by enhancing diversity in feature extraction, IEEE transactions on neural networks and learning systems 30.9 (2019): 2650-2661.
  • e) An embedded feature selection method for imbalanced data classification, IEEE/CAA J. Autom. Sinica, vol. 6, no. 3, pp. 703–715, May 2019.
  • f) Feature selection for multi-label classification using neighborhood preservation, IEEE/CAA J. Autom. Sinica, vol. 5, no. 1, pp. 320-330, Jan. 2018.

Author Response

Please see attachment for full letter of response to reviewers

Authors have provided below responses (in red) to the reviewers’ comments (in black and numbered). The manuscript has been edited using “track changes” as instructed by the email received from the Journal. In order to demonstrate or changes in text following reviewer comments, “comment” bubbles were added to describe the essential edits that were made a response to the reviewer recommendations below. Line numbers expressed here are referring to the edited manuscript with “show all markup” turned on.

Reviewer 1

In this paper, the authors propose a framework to extract the casual pathways through the casual relation determination especially for the Thai documents where the zero anaphora or the implicit noun phrase characteristics bring several problems. By using machine learning mechanism and linguistic phenomena, the proposed method determines CErel on each EDU-concept pair from the documents, extracts casual pathways from several EDU-concept pairs, and indicates the implicit mediators on the correct extracted causal pathways from the documents. Experimental results on documents show its high accurate extraction performance. However, I still have some comments for authors:

  1. Novelty and importance of target problem should be further illustrated that it has never been considered and addressed before.

    The novelty and importance of the target problem have been clarified in text following the recommendations provided by the reviewer (see lines 159 – 184). Authors have also provided additional text to explain the contributions, importance, and novelty of the study in lines 221-232 of the edited manuscript.

  2. The authors descript that the proposed framework has advantages in at least three different parts, but there are no more detailed information about that compared with other methods.

    See comment 1

  3. The author introduces a lot of examples to depict the process of the framework. But the detailed parts of examples are not classified in a proper way, maybe the author can provide a better way to describe.

    This description is provided at lines 366-373. Each CEpairI is CErel by Machine learning.  All CEpairI with CErel are collected into the WCP set. Furthermore, WCP is used to construct the causal pathway where the number of CEpairI depends the wrdCoc-pair matching on sliding a window size of two consecutive wrdCoc features.

  4. CEpairs has five pathways that are grouped into two kinds of different ways to express the example 5, and four pathways for example 6. How to determine the number of the pathways? And it is necessary for all the pathways?

    First of all, the examples orders have been remade. The previous “Example 5 and 6” is now “Example 4 and 5” respectively. 

    Regarding pathways, Figure 2 has One pathway consists of five events/states while Figure 3 have one pathway consists of four events/states. Each CEpairI is CErel by Machine learning before constructing the causal pathway.

  5. There are five steps in the proposed framework overviewed in figure 5, but the order and the novelty cannot be seen in it. Moreover, the description of figure 4 is not shown up completely that needs to place in a normal way. A same situation also emerges in table 1.

    Figure 5. Box 4.4 show Implicit-Mediator Indication by the TransCEPair set determined by the Transitive Closure And Box 4.1 show how to obtained wrdCoc.

  6. The author can read more recent studies regarding feature representation, sparse data analysis, and other related feature extraction techniques to improve the method of this paper. For instance, the following papers are highly relevant:

    Authors would like to express our gratitude to the reviewers for suggesting such literature. It was challenge to find literature since journal access was limited in our universities. We did manage to read the suggested materials below.

Authors went on to study the suggested literature by Leng et al (2019) and found it to be interesting. However, the differences between literature and this study are explained further at lines 229-232.

Authors went on to study the suggested literature by Ayinde et al (2019) and found it to be interesting. Unfortunately, as exemplified in the edited manuscript, we found that their approach was not compatible to our study. Explanations are further provided lines 173 to 177.

Reviewer 2 Report

The paper has serious type errors and for this reason it is very hard to read it. There are so many that I consider that the authors did not read the manuscript before send it and for this reason I ask them to fix all the errors such as: closed parenthesis without a pair one, unjustified spaces in text, texts that overlap other texts, etc. 

The Introduction section does not present the overall study and its connections. Instead it directly gives examples of the method usage and who they are handled. 

Then the method presentation becomes harder to be read as the figures overlap the texts.From the exemplifications I understand the cases presented but not how the system works in general, some structures must be provided but they are not sufficient. The method is anyway very laborious and without a proper presentation it is hard to read and understand it.

I strongly suggest the authors to improve the study presentation and to submit the paper again.

Author Response

Please see attachment for full letter of response to reviewers.

Authors have provided below responses (in red) to the reviewers’ comments (in black and numbered). The manuscript has been edited using “track changes” as instructed by the email received from the Journal. In order to demonstrate or changes in text following reviewer comments, “comment” bubbles were added to describe the essential edits that were made a response to the reviewer recommendations below. Line numbers expressed here are referring to the edited manuscript with “show all markup” turned on.

Reviewer 2

  1. The paper has serious type errors and for this reason it is very hard to read it. There are so many that I consider that the authors did not read the manuscript before send it and for this reason I ask them to fix all the errors such as: closed parenthesis without a pair one, unjustified spaces in text, texts that overlap other texts, etc. 

Authors would like to show sincere apologies. Authors have tested the document in various computers to check the formatting and edit. However, we do not have the latest Microsoft office versions. As a response to this comment, we have used a newer office version.

Moreover, for example, table 1 in line 629 will break if it moved. For the current version we therefore push it to the next page hoping this will not break for you.

Authors appreciate reviewer’s comment on the formatting issues as we would never realize this was a problem. We have attempt to fix all the formatting issues as a response to reviewers’ recommendations.

  1. The Introduction section does not present the overall study and its connections. Instead it directly gives examples of the method usage and who they are handled. 

    Authors have previously provided in the manuscript (lines 248-252) for the flow and structure of the presented study. Following the recommendations provided by reviewer we understand that the work being presented isn’t clear enough. Therefore, readjustments were made to bring clarity to the study.

Authors have attempted at restructuring and provided additional text for the introduction section (lines 59-254) as a response to reviewer’s recommendations. In order to present the overall study, their connections, and the contributions of the study, additional texts are given at lines 154-184 and lines 221-232. Since the paper is presented based on the engineering methodological approach being proposed, the structure of the paper follows the objective of the study. Therefore, our introduction section follows as such. Through restructuring the introduction with additional texts provided, authors hope to bring greater clarity and comprehension of the introduction section. Authors wishes to thank the recommendations provided by the reviewer.

  1. Then the method presentation becomes harder to be read as the figures overlap the texts.

    See response in comment 1.

  2. From the exemplifications I understand the cases presented but not how the system works in general, some structures must be provided but they are not sufficient.

    The examples have been restructured (e.g. lines 329-342) throughout and figures redrawn (e.g. line 801) in order to bring further clarity into how the system works.

  3. The method is anyway very laborious and without a proper presentation it is hard to read and understand it.

    Authors wishes to apologize for the lack of proper presentation. Authors therefore employed a more structured way of presentation following previous publications authors have made in the past.

  4. I strongly suggest the authors to improve the study presentation and to submit the paper again.

    Authors have attempted to improve the study presentation and would like to thank the reviewer for suggestions made. Authors have employed various experiences gained from past publications into this paper.

Round 2

Reviewer 2 Report

The article presentation still has problems, for example at line 604 there is a caption to a figure that is included in the next page.

Also, I consider that the whole presentation is too large and that it is requires a significant reading and understanding effort by the fact that the language in which is implemented is other than English and for this reason a lot of textual sequences does not bring any information for many readers. Also, there are many study-cases presented and in this manner the main structure of the system is not properly evidenced in the paper.

As a consequence, I still consider that this paper is a hard material and could benefit from a better presentation.

Author Response

please see attachment for the letter of response to reviewer. I have copied the text from the attached file and paste it below.

Response to Reviewer

Authors would like to show appreciation for the recommendations provided by the reviewer. Authors have attempted to make changes as a response. The attached file is made under “track changes” as instructed by the editor.  Comment box were added to direct both the reviewer and editor to exact locations where changes were made.

Below are the responses (in red) to each of the reviewer’s comment (in black).

  1. The article presentation still has problems, for example at line 604 there is a caption to a figure that is included in the next page.

    Authors could not locate the caption on line 604 specified by the reviewer. Below are screenshots for line 604 in our document both with and without markup

    << see in attached file for screenshots >>

    as you can see above, line 604 for the authors using MS Word 2016 is just text with no caption present. Is it possible if the reviewer can provide a screenshot of the dislocated caption? So authors can fix it.

  2. Also, I consider that the whole presentation is too large and that it is requires a significant reading and understanding effort by the fact that the language in which is implemented is other than English and for this reason a lot of textual sequences does not bring any information for many readers. Also, there are many study-cases presented and in this manner the main structure of the system is not properly evidenced in the paper.

    Authors agree to the reviewer and have attempted to resolve this issue. First, we try to reduce the number of examples and give more clarity to each example instead. For example, the “Example 1” in lines 149-173 of the “marked up” track change text was used to convey the pathway (lines 80-82) and the problems (lines 321-325 section 3.1 and lines 383-403 section 3.2). Example 1 was used to demonstrate the problem of some intervention of a non-causative/effect concept EDU(s) which may occur on the sequence of CEpairi on the document.

    Second, the systems overview of figure 5 was remade to improve the comprehension on the sequences for the readers (lines 572-573). Although, the language used in the study is not English, but the problems addressed and the method proposed are not limited by language. We emphasized this point at lines 1043-1046.
  3. As a consequence, I still consider that this paper is a hard material and could benefit from a better presentation.

    Authors have attempted to improve the presentation of the text as a result of reviewers recommendations.
